# Subjective Evaluation on the Annoyance of Environmental Noise Containing Low-Frequency Tonal Components

**DOI:** 10.3390/ijerph18137127

**Published:** 2021-07-03

**Authors:** Miki Yonemura, Hyojin Lee, Shinichi Sakamoto

**Affiliations:** Institute of Industrial Science, The University of Tokyo, Komaba 4-6-1, Meguro-ku, Tokyo 153-8505, Japan; leehj@iis.u-tokyo.ac.jp (H.L.); sakamo@iis.u-tokyo.ac.jp (S.S.)

**Keywords:** annoyance, environmental noise, tonal audibility, subjective evaluation

## Abstract

Recently in Japan, noises from wind turbines and domestic use heat sources sometimes cause an increase in noise annoyance owing to low-frequency tonal components. The purpose of this study was to investigate the effects of the tonal components on the annoyance of the environmental noise. The authors conducted an auditory test in the laboratory to evaluate the annoyance of tonal noise using a seven-step rating method. The stimuli were composed of a broadband noise modeling of the environmental noise (25, 30, and 35 dB) and a low-frequency tonal component. With the tonal component added to the broadband noise, the frequency and tonal audibility were varied to 40, 50, 100, 200, and 400 Hz and 0, 3, 6, 9, and 12 dB, respectively. The amount of increase in annoyance owing to the addition of the tonal component was quantitatively evaluated as a tonal adjustment by comparing it with broadband noise. As a result, tonal adjustment ranged from 0 to 7 dB, and the higher the tonal frequency, the larger the value. For the test background noise level, the lower the background noise level of the test sound, the greater the value. This trend suggests that the influence of tonal components on subjective impressions is stronger in quiet environments such as residential areas. This result may provide a basis for the evaluation method, which varies the penalty in the noise evaluation according to the frequency of the pure tones and the noise level.

## 1. Introduction

Tonal noise contains a prominent narrowband component. Machinery noise and wind turbine noise often include such tonal components. To evaluate the tonality of such noises, the tonal audibility (TA) has been defined, and international standards such as ISO/PAS20065 [1] and IEC61400-11:2012 [2] provide the calculation methods for the value of the index. Based on the index, noise assessment guidelines for wind turbine noise are regulated in several countries and regions, and a noise “penalty” is added to A-weighted or band levels to assess wind turbine noises [3,4]. For an assessment of wind turbine noise in Japan, the Ministry of the Environment of Japan has published a manual for the measurement of wind turbine noise in 2017 [5], and in the manual, to assess the wind turbine noise, a noise guideline based on A-weighted continuous-equivalent sound pressure level, *L*_Aeq_, was presented. Regarding the tonal component in the wind turbine noise, it was reported that some wind turbine noises include tonal components within the frequency range of 50–1000 Hz with a maximum TA level of approximately 15 dB [6]. In view of this fact, the tonal noise problem may also be problematic in Japan, and therefore the manual requires the measurement of TA according to the guidelines of foreign countries. However, the manual has not yet shown the assessment criteria. Further knowledge on the effects of tonal noise on humans is required in Japan. In addition to wind turbine noise, in Japan, the noise from residential heat pump water heaters has been identified as another problematic noise source, which generates prominent low-frequency tones, of which the tonal frequency may range from 25 to 40 Hz, and their harmonic components occasionally appear. There is a concern that such tonal noise may cause health problems by sleep disturbances. To cope with the problems caused by such tonal noises, in Japan, it is necessary to collect quantitative data with scientific support.

The problem associated with wind turbine noise has been a concern, and a dose–response relationship has been reported in social surveys. Pedersen and Persson Waye [7,8] conducted a social survey in several areas of Sweden and reported on the relationship between annoyance and lower sleep quality owing to wind turbine noise. Janssen et al. [9] also showed a dose–response relationship for annoyance based on the results of social surveys in Sweden and the Netherlands and compared it with other traffic noises. Kuwano et al. [10] conducted a social survey in Japan and reported the results of annoyance and effect on sleep disturbance. These studies have been referenced in the environmental noise guidelines by the WHO [11], and recommended values for *L*_den_ and *L*_night_ have been given in terms of annoyance and sleep disturbance. Haac et al. [12] recently conducted a nationwide survey in the U.S. and showed a dose–response relationship for annoyance.

The annoyance caused by tonal noise has been investigated by many researchers. Tonal noise was first focused on as a problem of aircraft noise, and was then treated as a problem of the working environment noises from equipment and machinery. Tonal noise has also been recently considered a characteristic of wind turbine noise. Due to the complexity of the frequency response of tonal noise, most studies have been conducted in laboratory experiments where the test sound is carefully controlled. Little [13] focused on aircraft noise and conducted experiments to evaluate the annoyance of jet engine noise and artificial noise with tonal components within the range of 850–6800 Hz. Based on the experimental results, the author proposed a penalty based on the tonal frequency and tonal level. Landström et al. [14] conducted evaluations of noise in offices, laboratories, and factories. A total of 439 cases of noise with *L*_A_ between 40 and 80 dB were evaluated for their effect on annoyance, and the results showed that the presence of tones increased the annoyance ratings. It was reported that the increase in annoyance owing to tones may correspond to a level difference of approximately 4 dB. In the ETSU report [15] on the rating of tonal noise from windfarms in the UK, subjective evaluation experiments were conducted to assess the audibility of pure tones, loudness, and the penalty in an annoyance evaluation for wind turbine noise. The tonal frequencies were between 200 and 2400 Hz. Both loudness and annoyance were examined, and it was reported that the penalty for annoyance was higher than the increase in loudness. Ryherd and Wang [16] studied the effect of tonal noise generated from mechanical systems in office buildings on human task performance and perception. They used tonal noise with 120, 235, or 595 Hz tonal components at one of two tonal prominence ratios as the test sound and found that the noise conditions with tonal prominence ratios of 9 dB were generally perceived to be more annoying than those with ratio of 5 dB, although statistically significant differences in task performance were not found. Oliva et al. [17] conducted evaluation experiments using low-level test sounds, assuming a residential environment. The frequency range of the tonal component was extended to the low-frequency side from 50 to 2100 Hz. Based on the results, Hongisto et al. [18] proposed a model equation fitting the penalty by tonal audibility and the frequency of a pure tone. Hansen et al. [19] investigated the perceived sleep acceptability of a wind turbine noise containing a 50 Hz amplitude-modulated tone through listening tests. They reported that an increase in the depth of amplitude modulation at a tonal audibility of 12 dB was associated with a lower acceptability for sleep in noise-sensitive individuals.

As is described above, although many researchers have reported evaluation experiments on tonal noises whose frequencies are in a relatively wide frequency range, but there have been few reports discussing tonal noise including low-frequency tones at below 100 Hz. The results of studies on tonal noise in other countries and the noise evaluation guidelines based on such studies are expected to be applicable to Japan. However, tonal noise in Japan is characterized by a low-frequency tonal component, which is often below 100 Hz, and by the fact that it causes sleep disturbances in a quiet environment at bedtime. To solve these problems, the following two points must be clarified:Psychological responses to noise, including low-frequency tones of below 100 Hz.Psychological responses to situations with low test background noise levels.

In addition to differences in the acoustic properties of noise, cross-cultural differences in the community response to noise annoyance have been reported [20]. Therefore, it is necessary to conduct evaluation experiments with Japanese subjects. As described above, the authors studied through laboratory experiments the psychological effects of the low-frequency tonal components, including frequencies of below 100 Hz, in Japanese subjects.

### 1.1. Laboratory Annoyance

The psychometric properties of noise can be divided into three attributes: loudness, noisiness, and annoyance. To solve the public noise problem, although annoyance of noise should be quantitatively studied, the measurement of annoyance in terms of its true meanings is not easy to conduct because annoyance is a complex sensory quantity that is affected by non-acoustic factors such as the meaning of the test sound and the listening situation [21]. Social surveys may be suitable to cope with the annoyance in actual life. However, it is difficult to control several parameters of noise and to determine the relationship between the physical parameters and the response. In this study, the auditory tests on tonal noises were performed in the laboratory; therefore, we conducted subjective experiments on the “laboratory annoyance”. Through experiments on laboratory annoyance, the control of the physical parameters is easy, and the relationship between the parameters and responses can be obtained; nevertheless, the non-acoustic factors cannot be fully discussed. Västfjäll [22] reported that mood introduction and noise sensitivity of participants affect the judgment of noise annoyance in laboratory experiments. In this study, we attempted to control the mood based on interviews regarding noise just prior to the experiment, whereas we did not control the noise sensitivity. The details of this are presented in Section 2.3.

### 1.2. Study Outline

The purpose of this study is to quantitatively understand the effect of low-frequency tonal components on laboratory annoyance under quiet environmental noise conditions. For this purpose, experiments to evaluate laboratory annoyance were conducted using test sounds consisting of broadband noise modeling environmental background noise and a single tone.

The test stimuli used in the experiments were set to simulate the wind turbine noise and/or the noise from heat pump water heaters heard in residential areas, which may be problematic in Japan. The tonal component has a frequency ranging from 40 to 400 Hz, and a tonal level ranging from 0 to 12 dB in TA. The tonal frequencies and TA were set based on the measurement data from a field survey [6]. The background noise levels were 25, 30, and 35 dB, which simulates indoor noise in quiet residential districts in Japan at night (25 dB and 30 dB) and at daytime (35 dB). 

From the experimental results, we examined whether there was a difference depending on the tonal frequency and background noise conditions. The results were compared with the penalties provided in the noise evaluation guidelines and previous studies.

## 2. Method

### 2.1. Experimental System

Auditory tests were conducted in an anechoic room with an air volume of 210 m^3^ (depth, 4.04 m; width, 6.86 m; and height, 7.60 m) at the Institute of Industrial Science, The University of Tokyo (Figure 1 and Figure 2). Glass wool with a thickness of 30 cm was installed as a sound-absorbing material, and the theoretical cutoff frequency of sound absorption was 250 Hz. Strictly speaking, the room is not an ideal anechoic chamber in the lower frequency range. The impulse response using the swept-sine method was measured for 40–200 Hz, and the reverberation time *T*_30_ was determined to be between 0.1 and 0.4 s (shown in Table 1). The average sound absorption coefficient $\bar{\alpha }$ (–) was calculated from the measured reverberation time *T* (s), surface area *S* (m^2^), and volume *V* (m^3^) of the room using Sabine’s equation:(1)$$T=0.161\frac{V}{S\bar{\alpha }}$$

To reproduce the test sounds, 16 woofers (Fostex, FW405N) and a squawker (Auratone, SUPER-SOUND-CUBE) were placed on the wall. The distance between the subject and the loudspeakers was 3.5 m. Since the interior of the room is not a simple pressure field but a standing-wave sound field within the low-frequency range, the listening point was carefully set considering the sound pressure distribution in the room [23]. To calibrate the frequency characteristics and the sound pressure levels of test sound, a sound level meter (Rion, NL-52) was placed at the midpoint between the subject’s ears under the condition without a subject.

The test sounds were generated using Cool Edit Pro. The sound signals were divided into two parts by low-pass and high-pass filters with a crossover frequency of 224 Hz and distributed to the two types of loudspeakers, woofers and a squawker. Before sending the woofers and a squawker, the signals pass the equalizing filters to flatten the frequency response at the listener position. The test signals (48 kHz, 32 bit) were played using the following systems: a Hewlett–Packard personal computer with an ASIO driver, a D/A converter (RME Fireface 802), and speaker amplifiers (Sony, SRP-P4005; Accuphase, Pro-30).

### 2.2. Test Stimuli

The test stimuli were composed of broadband noise and a pure tone. Detailed information on the components, i.e., the frequency characteristics of the broadband noise, the level of the broadband noise, the tonal frequency, and the tonal level are shown in Table 2.

Two types of broadband noise were adopted as the background noise: indoor and outdoor. For the outdoor situation, an artificially synthesized noise with a frequency characteristic having a slope of −4 dB/octave in the band was used, referring to the measurement results of wind turbine noise around wind turbines by Tachibana et al. [24]. The authors measured wind turbine noises at 164 points placed around 29 wind farms and found that almost all wind turbine noises have similar spectral characteristics approximated by a slope of −4 dB/octave in the band spectrum. For the indoor condition, the spectral characteristics of outdoor noise was attenuated by a house filter, which is a model of averaged sound insulation characteristics of a house expressed by the sound pressure level difference in a 1/3 octave band between indoor and outdoor. Tachibana et al. have proposed three types of house filters for a general detached house in Japan [25], and a house filter for a single-pane window with aluminum sash, as shown in Figure 3, was adopted in the present study.

The A-weighted sound pressure levels (*L_p_*_A_) of the modeled noise were adjusted to 25, 30, and 35 dB, assuming that they were heard indoors and outdoors, and a pure tone was added. The frequency characteristics of the noise are shown in Figure 4. Comparing the outdoor and indoor noises with the same *L_p_*_A_, the levels were almost the same within the frequency range of above 250 Hz, and the indoor noise had a higher energy within a low-frequency range.

As the tonal component included in the tonal noises, a pure tone was added to the synthesized noises, assuming the tonal noises generated from wind turbines and residential heat pump water heaters. The tonal level was set such that its TA provided in IEC 61400-11:2012 varied from 0 to 12 dB in 3 dB steps (with 5 steps in total). The frequencies of the pure tone were 40, 50, 100, 200, and 400 Hz.

The frequency characteristics of some examples of the test stimuli are shown in Figure 5. To investigate whether the effect of tones on the annoyance varies depending on the noise levels, the levels of background broadband noises were varied from 25 to 35 dB in steps of 5 dB, as shown in Table 2. These three steps of background noise levels represent the assumed situations of the subjects, i.e., 25 and 30 dB for bedtime, which is extremely (25 dB) and relatively (30 dB) quiet, and 35 dB for daytime where some noise occurs.

In addition to the tonal noises described above, as a reference, broadband noises with the same frequency characteristics but without tonal components were also used in the experiment. The A-weighted sound pressure levels of the broadband noise stimuli ranged from 20 to 45 dB in 5 dB steps. Annoyance evaluation results of the broadband noise stimuli were used as a “scale” for a quantitative evaluation of annoyance of the tonal noises. The number of test sound per subject was 58 (50 tonal noises and 8 reference noises) in total as shown in Table 2.

### 2.3. Procedure

The subjects were divided into three groups, A, B, and C, according to the level of the test sound, as shown in Table 2. For each group, 20 paid subjects ranging in age from 30 to 49 years with normal hearing capabilities participated in the experiment. Therefore, 60 subjects (30 males and 30 females) in total participated in this experiment.

Before starting the auditory tests, the hearing levels (HLs) of all subjects were measured, and all the subjects were confirmed to be less than 10 dB HL at frequencies of between 125 and 4000 Hz. To remind the participants of their typical living environment, as a mood introduction mentioned in Section 1.1, they were interviewed about what noises bothered them in their daily lives.

During the auditory tests, the subjects evaluated the annoyance of the test sound using a 7-step rating scale method. The subjects listened to the test sounds for 10 s under the assumption that they hear such sounds then try to sleep at home (Groups A and B) or relax in their living room (Group C). After the presentation of the sound, the subjects rated the annoyance of each sound on a scale from 1 (not at all annoying) to 7 (extremely annoying). The test sounds were randomly arranged and presented to the subjects three times. To ensure the reliability of the responses, the responses of the participants whose correlation coefficients for the three responses were 0.6 or higher were used for data analysis. As a result, data from 18 subjects in Group A, from 17 subjects in Group B, and from 16 subjects in Group C were analyzed to discuss the annoyance of the tonal noises. 

### 2.4. Statistical Analysis

The difference in annoyance owing to the tonal frequency was confirmed based on statistical tests. A two-way ANOVA was conducted on the responses to the test sounds with the same tonal level (TA = 0, 3, 6, 9, and 12 dB), using the tonal frequency and subject as factors. Significance was determined at the 5% level through multiple comparisons (Tukey’s HSD). R Studio software was used for the analysis.

## 3. Results

### 3.1. Relationships between Annoyance and TA

The mean values of the annoyance ratings of the tonal noise are shown in Figure 6. Overall, for all groups and for all tonal frequencies, the annoyance ratings increased with increasing TA. Multiple comparison tests (Tukey’s HSD) were used to compare the differences between the frequencies of the tonal components. Except for the indoor noises under 30 dB and 35 dB conditions (Figure 6e,f), significant differences owing to the tonal frequency were found (*p* < 0.05). At background noise levels of 25 dB, the annoyance ratings were higher in tonal noises containing 100, 200, and 400 Hz tonal components, as compared with 40 and 50 Hz. Even under outdoor noise conditions for 25 dB background noise (Figure 6a), significant differences among 100, 200, and 400 Hz were also observed. 

A comparison between the background noise (indoor or outdoor) showed that the indoor noise with 40 or 50 Hz tonal components resulted in a higher annoyance than the outdoor noise condition. This is probably due to the fact that the indoor noise had relatively higher energy within the low-frequency range, the tendency of which resulted from the spectral characteristics of the background noise attenuated by the house filter shown in Figure 3. In particular, the annoyance rating was remarkably low for tonal frequencies of 40 or 50 Hz under the outdoor noise condition with a level of 25 dB (low-level outdoor noise condition), as shown in Figure 6a. This was probably because the levels of the pure tone were equal to or lower than the absolute hearing threshold; therefore, the tonal components were too small to be perceived by the subjects.

### 3.2. Relationships between Annoyance and A-Weighted Sound Pressure Level

Figure 7 shows the relationships between A-weighted sound pressure levels and annoyance ratings. In Figure 6, the open circle shows the results of the test sound, which did not include the tonal component at all (referred to as “reference noise” in Table 2). For the reference noise stimuli, almost linear relationships between the A-weighted sound pressure levels and the values of the subjective annoyance ratings were found (R^2^ > 0.97), as shown by the regression lines (oblique solid lines) in Figure 7. Compared to the annoyance ratings for the reference noise (broadband noise), the annoyance ratings of all the tonal noises (indicated by the other symbols) were higher than the estimated annoyance rating values for broadband noise based on the regression lines at the same A-weighted sound pressure levels.

Using the linear relationship between the A-weighted sound pressure level and the subjective rating of annoyance for the reference noises, the increase in annoyance owing to the tonal components can be quantified in decibels. Such an analysis has been proposed and utilized in previous studies [14,17,18,26]. The quantification method will be discussed in the next section.

### 3.3. Quantification of the Increase of Annoyance Owing to Tonal Components

The experimental results described above showed that annoyance increased more as the tonal component became more intense. In this section, we quantified the increase of annoyance and compared it with the penalty values reported in other noise evaluation guidelines and research studies.

#### 3.3.1. Calculation of Tonal Adjustment

To quantitatively evaluate the increase in annoyance, we introduced the tonal adjustment, which translates the differences in psychological rating values into differences in the A-weighted sound pressure level subject to broadband environmental noise. The calculation method is shown in Figure 8. The regressed straight line for annoyance ratings on the reference noise stimuli (broadband noise simulating general environmental noise) was used to convert the psychological annoyance rating values into an A-weighted sound pressure level, *L** (dB). The tonal adjustment Δ*L* (dB) was defined as the difference between *L** (dB) and the A-weighted sound pressure level of the test sound, *L*s (dB) (Equation (2)).
(2)$$\Delta L=L^{*}-Ls$$

#### 3.3.2. Tonal Adjustment Obtained from the Experiment

Figure 9 shows the relationship between the TA and the tonal adjustment Δ*L* obtained from the experiment. As a reference, the tonal adjustment *k* provided in ISO1996-2: 2017 [3] is shown with a grey line. The tonal adjustment ranged from 0 to 7 dB. When the TA was approximately 6 dB or less, the tonal adjustment increased with the TA. However, as shown in Figure 9d–f, the tonal adjustment tended to remain flat or decrease when the TA was 9 dB or higher, especially for 40 and 50 Hz tonal frequencies. As one of the reasons for this trend, the A-weighted sound pressure level *L*s increased in the case of the indoor noise with low-frequency tonal components, as shown in Figure 7d–f.

With respect to the test background noise level, in Group C, in which the level of background noise was 35 dB, the value of tonal adjustment ranged from 0 to 5 dB. This was relatively lower than that of the other groups (Figure 9c,f).

The increase in annoyance owing to the addition of tonal components was almost the same for both types of background noise. However, the tonal frequency and the levels of the test sounds affected the tonal adjustment values.

## 4. Discussion

### 4.1. Comparison with Guidelines for Noise Evaluation

A tonal adjustment and similar correction values were proposed in various evaluation guidelines. In ISO1996-2:2017, the tonal adjustment value *k* was specified (shown in Figure 9). The tonal adjustment value ranged from 0 to 6 dB and increased as the TA increased. The results of our experiment showed a similar trend; however, Δ*L* was 2–4 dB higher than *k* shown in ISO1996-2:2017 when the tonal frequency was higher than 100 Hz. Although *k* is defined using only the TA as a parameter, the experimental results show that psychoacoustical responses to tonal noise differ according to tonal frequency and background noise. In our experimental results, the correction value Δ*L* tended to be higher when the tonal frequency was higher, or when the background noise level was lower. In particular, in the experiments assuming bedtime conditions (where the background noise levels were 25 and 30 dB), the tonal adjustment value obtained from the experimental results was occasionally up to 3 dB higher than *k*, suggesting that the adjustment values by ISO1996-2:2017 may underestimate the annoyance.

In addition to ISO1996-2:2017, several countries and states have established their own guidelines for evaluating environmental noise. The ETSU report [15] shows that there is a positive correlation between the TA and penalty values calculated by the joint Nordic method, and that 1 dB of TA corresponds to a 1 dB penalty. The guidelines for assessing wind turbine noise in Denmark, Norway, and some states in the U.S. stipulate that a penalty of 5 dB is added when the tonal component is clearly audible [4]. It can be stated that these existing guidelines and the results of this experiment are in a general agreement, despite some differences in the values.

### 4.2. Comparison with Previous Studies

Landström et al. [14] evaluated the annoyance of environmental noise in a working environment, and they reported that the annoyance of noise with tonal components at below 2000 Hz increased more than that of noise without tones, and the increase in annoyance was equivalent to 3–6 dB. The results of our study were similar. 

Brambilla et al. [26] conducted an annoyance evaluation experiment for tonal noise and impulsive noises. Although they did not control the tonal frequency or the tonal level, they reported that the penalty decreased by 0.24 dB with a 1 dB increase in the level of the test sound when comparing three levels of tonal sound: 45, 55, and 65 dB. Although the level of the test sounds differs, the relative relationship between the level of the test sound and the penalty is similar to our results. 

Oliva et al. [17] studied the penalty value in quiet situations, such as residential areas. They conducted an annoyance evaluation experiment for tonal frequencies ranging from 50 to 2100 Hz. They reported that the penalty was negative at low frequencies of below 100 Hz, while our results show that the tonal adjustment was greater than 0 dB. However, it is difficult to compare their results with our own because of the different test tone settings, target frequencies, TA, and in particular, the spectral characteristics of the background noise. With regard to the effect of the level of the test sounds, however, the results of the comparison at between 25 and 35 dB showed that the penalty was higher for the former, and the trend was similar to our results.

### 4.3. Limitations

Subjective experiments were conducted to evaluate the laboratory annoyance of simulated environmental noise including low-frequency tones, and the influence of tonal components on annoyance was examined. As a result, it was found that the amount of increase in annoyance differed depending on the level of the test sound and the level and frequency of the tonal components. The increase in annoyance was quantified, and its validity was discussed by comparing it with the existing evaluation guidelines.

Although these findings may contribute to the development of a more detailed evaluation method that reflects the acoustic characteristics of the test sound in a tonal noise evaluation, there is room for further discussion on the following points.

Frequency range of the tonal components: Since the purpose of this study was to examine in detail the low-frequency range, which is a problem in Japan, and for which there is little previous data, we conducted an experiment focusing on low-frequency tones. However, it is difficult to compare the results because the frequency range is lower than that covered in previous studies. In the future, it will be necessary to investigate the tonal component at higher frequencies of up to 2 kHz using the same experimental method.

Harmonics: To simplify the conditions, the pure tone to be added was singular; however, actual tonal noises often include some harmonics and overtones owing to the noise generation mechanism. Since it has been reported that the subjective impressions of noises with harmonics are different from those when a single tone is included [27,28], the annoyance is also expected to be affected.

Control of the situation and traits of participants: The participants were selected from the public outside the university, and it was believed that the results obtained in this study are representative of the average response to noise. However, there is a possibility that the judgment of annoyance was affected by the participant’s trait, such as noise sensitivity, the living environment, and age. These issues need to be investigated further.

## 5. Conclusions

We investigated the influence of the tonal component within the low-frequency range of 40–400 Hz on the annoyance of environmental noise in a laboratory experiment. As the novelty of our study, we focused on the tonal component within the low-frequency range and we used test tones that simulate the level and frequency characteristics of quiet indoor and outdoor noise, and found the following results.
The addition of the tonal component increased the annoyance. The amount of increase depended on the tonal level, tonal frequency, and the level of the background noise. The increase in annoyance was greater when the background noise was quiet (25 and 30 dB). The higher the tonal frequency is, the higher the annoyance.The increase in annoyance owing to the tonal component was quantified and expressed as tonal adjustment Δ*L*. The tonal adjustment values were 0–7 dB for the 25 and 30 dB conditions (bedtime) and 0–5 dB for the 35 dB condition (daytime), which is comparable to the correction value *k* given in ISO1996-2:2017.As a feature of the low-frequency range, the annoyance tended to be lower when the overall noise level was low. This may be due to the fact that it is difficult to hear the tone when the energy of the tonal component is close to the absolute threshold of hearing, even if the TA, which is the relative energy ratio of pure tone and noise, is the same.


Although there is still much room for further investigation, as discussed in Section 4.3, the results of this study will contribute to the development of more detailed and appropriate evaluation methods for tonal noises, using the level of background noise, tonal frequency, and TA as parameters.

## Figures and Tables

**Figure 1 ijerph-18-07127-f001:**
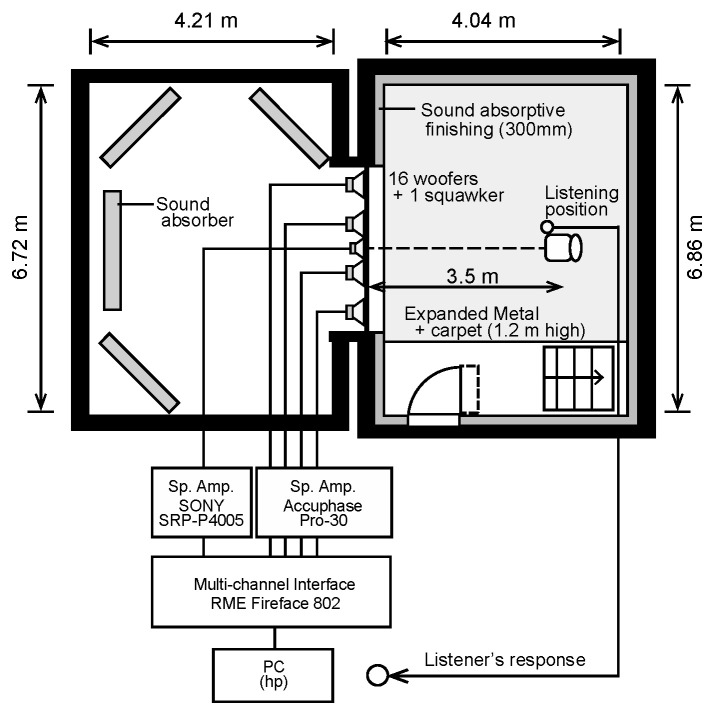
Sound reproduction system.

**Figure 2 ijerph-18-07127-f002:**
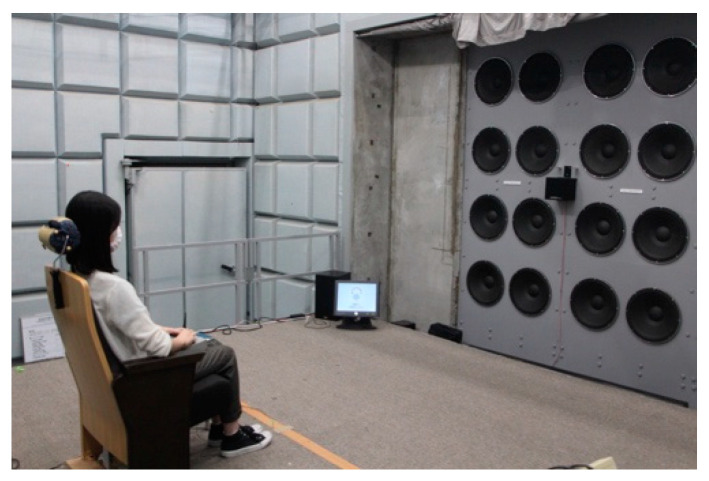
Speaker system in the anechoic room.

**Figure 3 ijerph-18-07127-f003:**
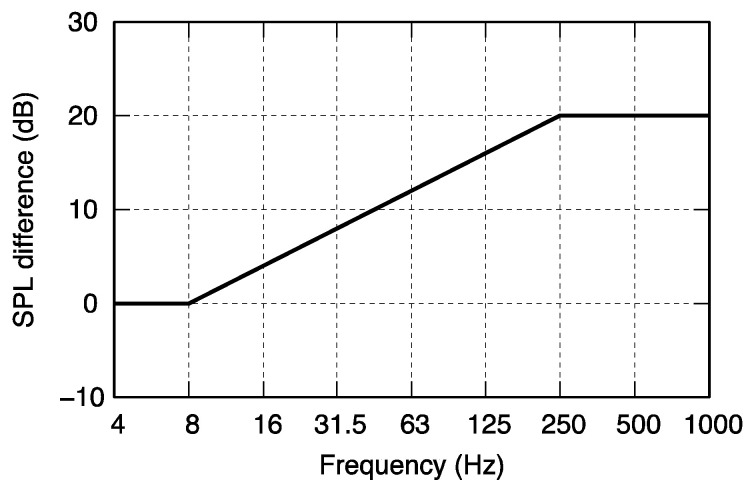
House filter for a single-pane window [25].

**Figure 4 ijerph-18-07127-f004:**
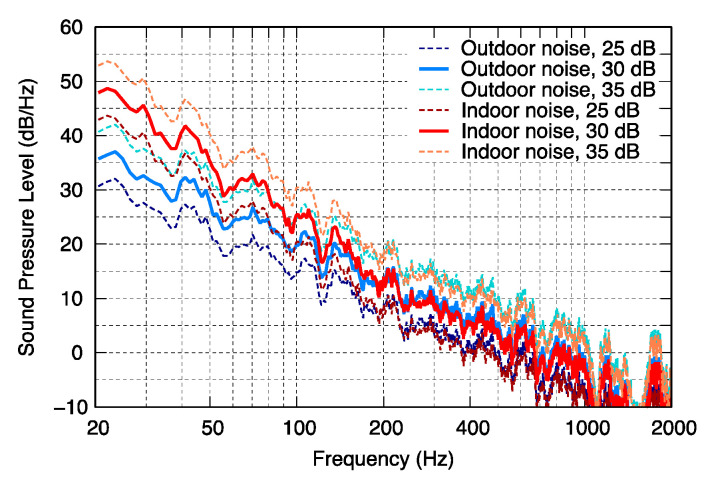
Modeled environmental noise used for test sounds.

**Figure 5 ijerph-18-07127-f005:**
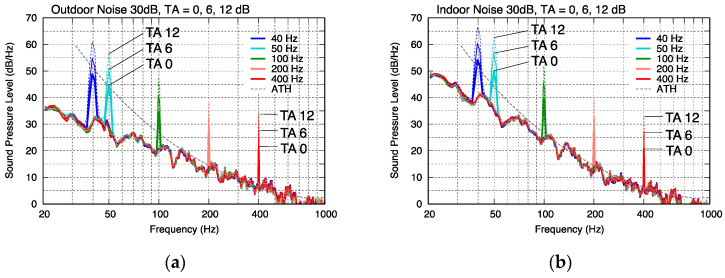
Frequency characteristics of test stimuli composed of broadband noise with an A-weighted sound pressure level of 30 dB and a synthesized tonal component at a TA of 0, 6, and 12 dB. (**a**) Outdoor noise (−4 dB/octave). (**b**) Indoor noise (−4 dB/octave. + house filter).

**Figure 6 ijerph-18-07127-f006:**
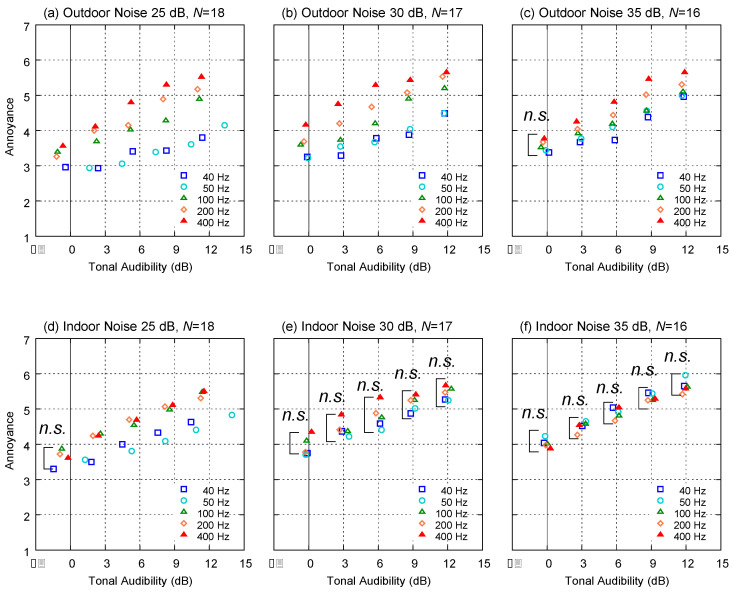
Relationships between tonal audibility (TA) in IEC61400-11:2012 and rated annoyance. Those for which no significant differences based on frequency were found through multiple comparisons are indicated as “*n.s.*” (**a**,**d**): Average values of Group A; (**b**,**e**): Group B; (**c**,**f**) Group C.

**Figure 7 ijerph-18-07127-f007:**
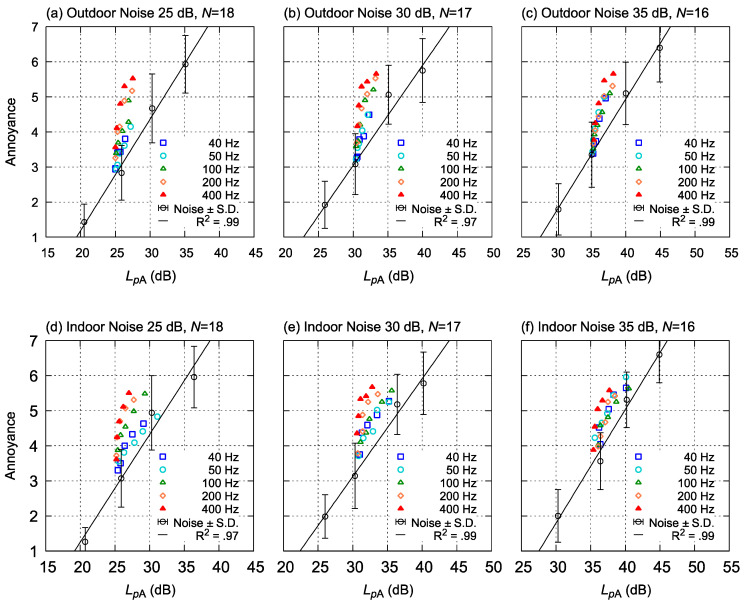
Relationships between A-weighted sound pressure level (*L_p_*_A_) and rated annoyance. Annoyance of noise without tonal components is shown in a white circle, and a strong linearity was observed between *L_p_*_A_ and annoyance. Colored plots show the result for tonal noise. (**a**,**d**): Average values of Group A; (**b**,**e**): Group B; (**c**,**f**) Group C.

**Figure 8 ijerph-18-07127-f008:**
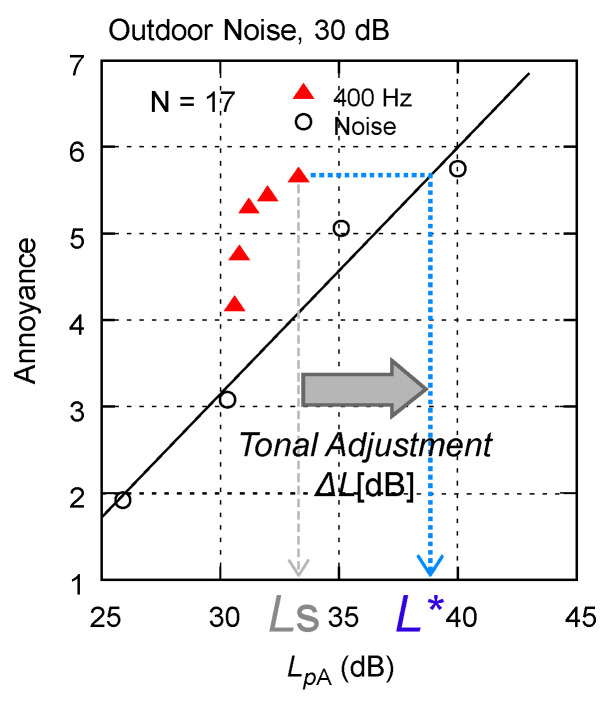
Calculation of tonal adjustment Δ*L*.

**Figure 9 ijerph-18-07127-f009:**
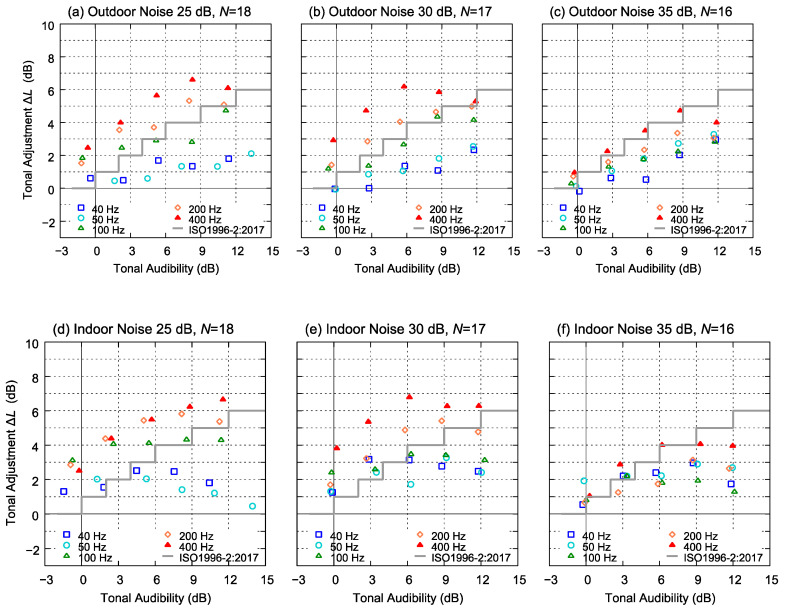
Tonal adjustment Δ*L* obtained from the experiment as a function of TA and the frequency of the tonal components. (**a**,**d**): Average values of Group A; (**b**,**e**): Group B; (**c**,**f**) Group C.

**Table 1 ijerph-18-07127-t001:** Reverberation time (*T*_30_) and average sound absorption coefficient ($\bar{\alpha }$ ) measured in the anechoic room.

	40 Hz	50 Hz	63 Hz	80 Hz	100 Hz	125 Hz	160 Hz	200 Hz
*T*_30_ (s)	0.30	0.27	0.25	0.38	0.11	0.14	0.10	0.15
$\bar{\alpha }$ (–)	0.52	0.57	0.63	0.41	1.35	1.07	1.60	1.02

**Table 2 ijerph-18-07127-t002:** Test sound. The test background noise levels of test sound were varied by 5 dB steps for 3 groups. The test background noise levels for Group A (C) were 5 dB lower (higher) than that of Group B.

Group	Assumed Situation	Type of Test Sound	Background Broadband Noise		Tonal Frequency	Tonal Audibility	Number of Test Sound per Subject
			Spectrum	*L_p_* _A_			
Group A	Trying to sleep at home	Tonal	indoor/outdoor	25 dB	40, 50, 100, 200, 400 Hz	0, 3, 6, 9, 12 dB	50 + 8
		Reference	indoor/outdoor	20, 25, 30, 35 dB	-	-	
Group B	Trying to sleep at home	Tonal	indoor/outdoor	30 dB	40, 50, 100, 200, 400 Hz	0, 3, 6, 9, 12 dB	50 + 8
		Reference	indoor/outdoor	25, 30, 35, 40 dB	-	-	
Group C	Relaxing in the living room	Tonal	indoor/outdoor	35 dB	40, 50, 100, 200, 400 Hz	0, 3, 6, 9, 12 dB	50 +8
		Reference	indoor/outdoor	30, 35, 40, 45 dB	-	-	

## Data Availability

The data presented in this study are available on request from the corresponding author. The data are not publicly available due to their containing information that could compromise the privacy of research participants.

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
