# Peer review of "Subjective Evaluation on the Annoyance of Environmental Noise Containing Low-Frequency Tonal Components"

_ijerph, 2021, doi:10.3390/ijerph18137127_

Round 1

Reviewer 1 Report

General remarks

The paper is interesting and useful.

I would suggest a thorough editing of the text. Moreover, it would be better to focus on the impact of the research in international level.

I would suggest to the author to provide a brief list of the quantitative outcomes of the research accompanied with a brief analysis in one of the sections 4.3, 4. 4 or 5  

Author Response

We are grateful to the reviewers for their thoughtful and constructive comments on our paper. We have addressed all the comments to the best of our ability. We have been able to incorporate changes to reflect most of the suggestions provided by the reviewers. We have highlighted the changes within the manuscript. The major changes are as follows.

1- In the introduction section, the novelty of our study has been clarified by explanations in lines 95-111.

2- In method section, the reverberation time of the anechoic room were measured and the results were added to Table 1. This showed that the properties of the experimental room were sufficient (lines 147-179).

3- In sections 3 to 5, we have changed the section structure based on the editor's comments:

The tonal adjustment, which was mentioned in Discussion section, was moved to Section 3.

The limitations of research, which was mentioned in the latter half of Conclusion, was moved to a separate section 4.3, and discussed in detail.

4- Descriptions of author contributions, research ethics, and conflicts of interest have been added as back matters.

The responses to the reviewers are attached in separate files.

In addition to the above changes and comments, all spelling and grammatical errors were checked by an English editing service.

We look forward to hearing from you regarding our submission. We would be glad to respond to any further questions and comments that you may have.

Sincerely,

Miki Yonemura, PhD

Research Associate

The Institute of Industrial Science, The University of Tokyo

4-6-1 Komaba, Meguro-ku, Tokyo, Japan 153-8505

+81-3-5452-6425

m-yone@iis.u-tokyo.ac.jp

Reviewer 2 Report

This paper presents a subjective evaluation of the annoyance from environmental noise that contains low frequency tonal components. The paper reads well in most part and the work is interesting. I have the following comments for the authors to address:

1- There are similar work in the literature, as cited properly in the introduction. However, the justification for this work is not clear. The characteristics of environmental noise would be dependent on the type of equipment that contribute to such noise but it is not dependent on the country. So, if similar work is done in another country it should be applied to Japan. Please clarify the novelty of your work.

2- Has there been a control sample from the sixty subjects that have participated in this experiment?

3- Tonal noise generated from wind turbines has a much lower frequency than the tonal noise components that were introduced in the test. 40 to 400 Hz are way higher than the tonal frequency expected from wind turbines (typically below 10 Hz). Can the authors clarify why is that?

4- Human hearing mechanism (i.e. the threshold of hearing and its frequency range) is known to change with age. Therefore, the perception of noise would vary from one person to another depending on how old they are. A classification of the sixty subjects that participated in the experiments based on their gender is not enough. Another classification based on their age must be added.

Author Response

(The authors gave the same response as above.)

Reviewer 3 Report

Dear editor, dear authors;

This work examines the relationship between tonal noise metrics and the perception of annoyance in communities. The focus of attention is placed on certain noise sources such as wind turbines and domestic HVAC (residential heat pump water heaters). The final objective is to adjust the penalties on the global value of LAeq to obtain the noise rating that better suits the real annoyance reached by the people due to the effect of an audible tonal component emerging from broadband noise. Since noise annoyance is subjective in nature, the authors conduct an auditory trial in Japan as a precursor study that will inspire future guidelines in Japan that objectively measure the prominence of the tones to finally assess the noise annoyance. These tests cover five tones between 40Hz and 400Hz and five energy levels emerging (TA) over the corresponding band of background noise. The broadband (background) noise is scaled in three steps but using outdoor artificial noise (simulating an environmental noise whose spectrum is a representation of a generic windfarm) and its indoor replica (house filtering). If I'm not mistaken, this does 150 tonal different kinds of tests (indoor + outdoor, as it is shown in Figure 5 and Table 1) + 12 broadband tests alone (because the indoor-outdoor filtering difference, as it is shown in Figure 6 and Table 1). The test is carried out for sixty subjects in an anechoic chamber (or semianechoic?). Any reader of the journal who is in a hurry and begins reading with the results section finds some very explanatory and easy-to-understand graphics (5 and 6) at a glance. The explanations of the statistical tests and some previous interpretations of the results are easy to follow. This is followed by a relevant and meaningful discussion section that introduces and explores some interesting insights. The comparisons of results with different guides and previous works are very interesting, especially the deviation from the ISO-1996:2 recommendations. Up to here and apparently a solid study. However, the conclusions appear to be a summary of the work and do not concentrate on the main research findings. In any case, this would be easy to solve. In my opinion, the limitations of the study should be removed from the conclusions and included in a separate section. We will talk about these limitations later.

But, after the reading of the paper with the code: IJERPH- 1259304, and titled: Subjective evaluation on annoyance of environmental noise containing low-frequency tonal components; my recommendation is to publish it with major corrections.

in general terms, it can be said from the article that the title and abstract are OK and anticipate what the paper is about. Some keywords are repetitive. The aim and the scope of the study are defined explicitly and are achievable. The justification of the study can be found in the conclusions: [328-330] “Although there is still much room for further investigations, the results of this study will contribute to the development of more appropriate evaluation methods for tonal noises”. Also, the structure of the paper is correct and the contents conveniently exposed. The English used is easy to follow although some errors appear, for example in the lines [53, 96, 128, 241, 301, 319] it can be read: “intensity of the tonal components” (authors should consider replacing “tonal intensity” by tonal energy or tonal level)

From here it will be explained what aspects of the work generate doubts and must be corrected or improved. Some aspects generate the greatest concern and that is why the necessity of "major corrections" has been recommended.

  1. Appropriateness of the state of the art revision and references.

The introduction doesn't incorporate the necessary references to describe the scientific environment in which the research is carried out. Only 12 references of which 4 are guides and standards support the background to be addressed in the introduction. The references cited in the text are not very current (only 4 of the 15 references were written in the last five years) and some hot issues are not reviewed (as we will see next). On the positive side references cited in the text support the justification of some decisions adopted in the study (objectives, methodologies,….). References that introduce the problem of the health of tonal noise in sleep are missed. Although the article focuses on tonality, perhaps it would be of interest to mention some studies on wind turbine noise and its effects in communities. An example of a macro-study supported by 1043 respondents:

- T. Ryan Haac, Kenneth Kaliski, Matthew Landis, Ben Hoen, Joseph Rand, Jeremy Firestone, Debi Elliott, Gundula Hübner, and Johannes Pohl. Wind turbine audibility and noise annoyance in a national U.S. survey: Individual perception and influencing factors The Journal of the Acoustical Society of America 146, 1124 (2019);

Continuing with this topic, one part of the evaluation is based on measuring the response of a group of people who try to sleep at home. The next paper focuses on some acoustic characteristics of turbine noise and its effects on sleep disturbance.

- Kristy Hansen, Phuc Nguyen, Branko Zajamsek, Gorica Micic(1), Peter Catcheside. Pilot study on perceived sleep acceptability of low-frequency, amplitude modulated tonal noise. Proceedings of the 23rd International Congress on Acoustics, 9–13 September 2019, Aachen, Germany: pages 1447–1454

Related to sleep disturbance also, no reference is made to previous studies that assessed this situation in anechoic chamber tests.

There is another topic that is worrying and should be considered in the introduction, and it is that of the possible differences in the assessment of annoyance due to cultural and social factors. For example in the study :

- T.Yano, T.Sato, M.Björkman, R.Rylander. Comparison of community response to road traffic noise in Japan and Sweden – part II: path analysis. J Sound Vib, 250 (2002), pp. 169-174

It is suggested that different customs of the people living in the two different countries and other social factors can influence annoyance.

About the authors claim:

[84-86] By the experiments on laboratory annoyance, the control of the physical parameters is easy and the relationship between the parameters and responses can be obtained, nevertheless, the non-acoustic factors cannot be discussed.

The experiments designed and carried out by the authors are important and legitimate for the construction of knowledge about the response of the human being to the impact of tonal noise. Why the non-acoustic factors cannot be discussed? Perhaps the authors meant: will not be discussed in this text. But there is a "non-acoustic" factor that should at least be taken into account in this test and that is personal variables such as sensitivity and mood. For example:

- D. Vastfjall

Influences of current mood and noise sensitivity on judgments of noise annoyance

J Psychol, 136 (2002), pp. 357-370

Which main conclusión suggest that people’s current states  (mood)  and traits (noise sensitivity) may have a joint effect on judgments of noise annoyance. Therefore, it appears to be important to control them in both laboratory and field studies of noise annoyance. This could be relevant, especially in a study where the number of subjects is not too large.

There is also no mention of works that have addressed noise problems specifically caused by HVAC equipment installed in residential buildings. So, the references cited in the text do not conveniently review the burning issues, The authors should conveniently discuss the scientific context in which their research is conducted and make a sufficient review of the state of the art on the topic addressed.

  1. Method.

Given the dimensions of the anechoic chamber, it would be interesting to know its design characteristics, in order to know the response of the chamber at low frequencies. What is the cutoff frequency of the chamber? What characteristics of absorption exhibit the chamber at the low frequencies that are being tested (at least at 40, 50, 100Hz)? It is not just a question related to the calibration of the sound pressure levels at a certain frequency during the test (with a sound level meter), it is a question related to the comparability of the different test conditions. For example, does the chamber guarantee that the reverberation affecting all the test tones is the same?

I miss section 2.4 dedicated to the description of the statistical tests that have been used.

Not all the theoretical and experimental framework is developed logically and understandably. Although the scope of the work focuses on the frequencies that are normally reported in wind turbines and domestic heat pump water heaters systems (HVAC) in Japan, I wonder why the authors have not extended the frequency band in the analysis, which would have been very interesting to extend the comparisons with guides and previous works (and could alert about confidence to the results in the anechoic chamber). Table 3 is not clear and needs to be revised. At least I don't understand anything about the distribution of test types between the three groups. Referring to figure 4. Are you sure that graphs 4.a. (outdoor noise) and 4.b. (indoor noise) show the same level of the background noise of 30 dB?

  1. Limitations of the study are NOT adequately discussed and how they can affect the conclusions

Regarding whether the research results are significant and generalizable, is it worth the authors to discuss some things:

- if the total number of subjects (18, 17, and 16) within the experiment allows the results to be representative

- If the characteristics of the chamber allow a comparison between the results at different frequency bands

- whether failing to control the mood and trait of the people being tested affects the quality of the results

- In which way the harmonics are affecting the auditory impression.

  1. Ethics

Although the trial does not appear to pose any risk to the subjects, it lacks approval from an independent body of collection and handling of data and the necessity of participants' informed consent to participate in the study.

Author Response

(The authors gave the same response as above.)

Round 2

Reviewer 3 Report

Dear editor, dear authors;

After the reading of the improved version of the paper with the code: IJERPH- 1259304, and titled: Subjective evaluation on annoyance of environmental noise containing low-frequency tonal components; my recommendation is to publish it with some minor corrections. I would like to congratulate the authors for the work and the improvements of this second version. So in this second round of review, my comments aim to contribute as much as possible to the clarity of the presentation and the contents. That is why I am going to skip the comparative description of the second version with respect to the first, to go directly to the doubts that this reviewer still has. Doubts that I want to discuss with the authors and I would greatly appreciate your comments and arguments, both for and against my views.

  1. [20, 105, 232, 244, 252, 253, 338, 380, 388, 406, 434] I would like to make sure of the meaning of presentation level". As I read the new corrections, doubts assail me. Do not authors believe that it would be more appropriate to say "background sound presentation level" (or "test background noise level") instead of "presentation level"?

  1. Please, check if the following title for the article would be more correct

"Subjective evaluation on the annoyance of environmental noise containing low-frequency tonal components"

  1. When you read the new section (4.3) on strength and limitations ..... don't you think the strengths are really the conclusions of the study? Taking into account that the aim of the study is (literally) “The purpose of this study is to quantitatively understand the effect of low-frequency tonal components on laboratory annoyance under quiet environmental noise conditions…..” The 3 strengths seem rather the answer to the objectives set.

  1. I cannot understand how many different types of tests the subjects in groups A, B and C. have undergone. I still have trouble understanding the distribution of the test to the 3 groups. If we join tables 2 and 3 into a single table, in order to group the tests, what would be the correct version of the grouped table? (I suggest 3 versions; please see the attached PDF file ) None of them?                                                                                                                                           I think there is something wrong with those tables. Please check. Is it correct (table 2) that the tests with "reference noise" are indoor and also outdoor? Is it correct (table 3) that 25, 30 and 35 dB refer to tonal noise? or does it refer to the level of background noise in the tonal tests?

Author Response

We are grateful to you for thoughtful and constructive comments on our paper. We have been able to incorporate changes to reflect all suggestions provided by the reviewer. The changes from the 2nd version of manuscript were highlighted in blue.

The major changes are as follows.

- The first half of the section 4.3 in second version, which was a summary of the results, has been moved to Conclusion. We have changed the title of 4.3 to “limitations.”

- Table 2 and 3 in second version have been combined into Table 2 in the latest version.

The responses to the reviewer 3 are attached in separate files.
